# Phosphorylation disrupts long-distance electron transport in cytochrome c

Alexandre M. J. Gomila[1,2,9], Gonzalo Pérez-Mejías [3,9], Alba Nin-Hill [4,9], Alejandra Guerra-Castellano[3], Laura Casas-Ferrer[1,8], Sthefany Ortiz-Tescari[1], Antonio Díaz-Quintana [3], Josep Samitier[1,2,5], Carme Rovira [4,6] ✉, Miguel A. De la Rosa [3], Irene Díaz-Moreno [3] ✉, Pau Gorostiza [1,2,6] ✉, Marina I. Giannotti [1,2,7] ✉ & Anna Lagunas [1,2] ✉

It has been recently shown that electron transfer between mitochondrial cytochrome $c$ and the cytochrome $c_1$ subunit of the cytochrome $bc_1$ can proceed at long-distance through the aqueous solution. Cytochrome $c$ is thought to adjust its activity by changing the affinity for its partners via Tyr48 phosphorylation, but it is unknown how it impacts the nanoscopic environment, interaction forces, and long-range electron transfer. Here, we constrain the orientation and separation between cytochrome $c_1$ and cytochrome $c$ or the phosphomimetic Y48$p$CMF cytochrome $c$, and deploy an array of single-molecule, bulk, and computational methods to investigate the molecular mechanism of electron transfer regulation by cytochrome $c$ phosphorylation. We demonstrate that phosphorylation impairs long-range electron transfer, shortens the long-distance charge conduit between the partners, strengthens their interaction, and departs it from equilibrium. These results unveil a nanoscopic view of the interaction between redox protein partners in electron transport chains and its mechanisms of regulation.

Many electron transfer (ET) reactions occurring in vivo implicate pairs of proteins in which at least one of them is free to diffuse, like the small electron-carrying proteins plastocyanin, ferredoxin or cytochrome $c$ (C$c$)[1]. Specific recognition and binding between proteins occur widely in biology. To reconcile an overall sustained and efficient ET process demands a high turnover rate. Therefore, the complex formed between redox proteins in the electron transport chain (ETC) must be transient, and balance specificity and binding strength[2,3]. The recent direct experimental observation of electrochemically gated, long-distance ET between cytochrome $c_1$ (C$c_1$, subunit of the cytochrome

$bc_1$) and C$c$ through the aqueous solution suggested that such tradeoff is achieved without direct contact, by molding the ionic distribution and electric field between their redox-active sites[4]. Thus, these redox protein partners might conciliate high specificity with weak binding, to keep the high turnover rate required by their biological function[5].

C$c$ is a mitochondrial hemeprotein that exerts multiple functions, and its location depends on the cell condition[6-10]. Under homeostasis, C$c$ localizes in the intermembrane mitochondrial space, transferring electrons from cytochrome $bc_1$ complex (C$bc_1$, complex III) to cytochrome $c$ oxidase (C$c$O, complex IV) in the ETC. Two binding sites for

[1]Institute for Bioengineering of Catalonia (IBEC), The Barcelona Institute for Science and Technology (BIST), Barcelona, Spain. [2]CIBER-BBN, ISCIII, Barcelona, Spain. [3]Institute for Chemical Research–cicCartuja, Universidad de Sevilla, Consejo Superior de Investigaciones Científicas (CSIC), Sevilla, Spain. [4]University of Barcelona, Department of Inorganic and Organic Chemistry, Institute of Theoretical Chemistry (IQTCUB), Barcelona, Spain. [5]Department of Electronics and Biomedical Engineering, University of Barcelona (UB), Faculty of Physics, Barcelona, Spain. [6]Catalan Institution for Research and Advanced Studies (ICREA), Barcelona, Spain. [7]Department of Materials Science and Physical Chemistry, University of Barcelona (UB), Faculty of Chemistry, Barcelona, Spain. [8]Present address: Laboratoire Charles Coulomb (L2C), UMR 5221 CNRS-Université de Montpellier, Montpellier, France. [9]These authors contributed equally: Alexandre M. J. Gomila, Gonzalo Pérez-Mejías, Alba Nin-Hill. ✉e-mail: c.rovira@ub.edu; idiazmoreno@us.es; pau@icrea.cat; migiannotti@ibecbarcelona.eu; alagunas@ibecbarcelona.eu

Cc on $Cc_1$ and CcO have recently been reported, a functional site (*proximal* site) enabling the correct orientation of redox centers to optimize the ET route, and a non-productive one (the so-called *distal* site) which is implicated in the turnover of Cc molecules[11,12]. Cc also takes part in other redox reactions within mitochondria, including reactive oxygen species (ROS) scavenging, redox-coupled protein import via Erv1-Mia40 pathway, and reduction of p66Shc generating ROS[8]. Upon distinct stimuli, such as oxidative stress or DNA damage, Cc can relocate to several cell organelles, interacting with proteins governing cell life and death[13–20]. Regarding oxidative stress, around 0.2-2% of the electrons involved in the ETC yield ROS via complex I (mainly), II, and III[21]. ROS have a dual role in cell signaling showing regulatory behavior at low levels and inducing apoptosis through oxidative stress at high ROS concentration. When mitochondrial production of hydrogen peroxide increases, Cc functions as a proapoptotic protein via its pseudo-peroxidase activity[22]. Around 15% of the Cc population is anchored to the inner mitochondrial membrane through binding to the lipid cardiolipin[23,24]. The pseudo-peroxidase function of the cardiolipin-Cc complex is activated at high levels of hydrogen peroxide and leads to the release of the hemeprotein from the mitochondria into the cytosol where it triggers apoptosis[25,26]. In mammals, Cc interacts in the cytosol with the apoptosis protease activating factor-1 protein (Apaf-1), to form the apoptosome and leading to downstream caspase activation and cell death[8,27,28].

These moonlighting Cc activities are tightly regulated by post-translational modifications (PTM)[29,30]. Notably, phosphorylation occurs in physiological and disease conditions and is highly tissue specific. Up to five tissue-specific phosphorylation sites of Cc have been characterized in mammals, including Thr28, Ser47, Tyr48, Thr58 and Tyr97[31]. Phosphorylation of Cc has been described to exert a cytoprotective and antiapoptotic role by downregulating ETC flux, thereby preventing mitochondrial membrane potential hyperpolarization, decreasing ROS production, and avoiding caspase activation. Phosphorylation of Cc has also been related to several pathologies, such as ischemia/reperfusion injury, neurological disorders, and cancer progression[32]. Although phosphorylation mainly occurs on serine and threonine residues with high frequency, phosphorylation of tyrosine residues has a crucial role in the regulation of mechanisms directly relevant to mitochondrial and cancer signaling, such as metabolic homeostasis, differentiation, and proliferation[33,34]. Among the two Cc tyrosine residues that can be phosphorylated in vivo, Tyr48 residue is located at the Ω-loop in the vicinity of the heme cleft, whereas Tyr97 lays far from the heme cleft in the C-terminal α helix of Cc. Cc phosphorylated on residue Tyr48 was primarily isolated from bovine liver tissue and it has been related to cancer[35]. Moreover, it has been found that Tyr48 can be mutated by histidine, which causes a mild human disease—thrombocytopenia—that results in a lower level of blood platelets[36]. Therefore, discerning the molecular mechanisms that regulate the function of phosphorylated Cc at position 48 is of great interest for understanding the role of Cc in pathology.

Functional studies revealed that Tyr48-phosphorylated Cc preserves the heme environment (no changes are observed in the Fe-Met80 absorption band at 695 nm) and that both Tyr48-phosphorylated and nonphosphorylated Cc species produced a hyperbolic kinetic response and the maximal turnover number of non-phosphorylated Cc was two times higher than that of phosphorylated Cc ($3.7 \, s^{-1}$ and $8.2 \, s^{-1}$, respectively)[35]. The technical difficulties to preserve the in vivo protein phosphorylation after purification, the large amount of phosphoprotein needed for structural and biophysical assays and the lack of information on specific Cc-Tyr48 kinases requires adopting other approaches to study this phosphorylated protein. Over the last years, some authors have developed engineered aminoacyl-tRNA synthetase systems that can incorporate O-phosphotyrosine. However, this approach presents multiple challenges as phosphotyrosine poorly penetrates the cells

due to its charge, it is unstable inside the cells and the incorporation efficiency by the aminoacyl-tRNA synthase is relatively low, producing a low yield of labelling (<5%) that is incompatible with single molecules techniques[37]. A convenient approach is using phosphomimetic mutants for functional studies[38]. The Cc Y48E mutant, in which Glu mimics the negative charge of the phosphate group, showed similar reaction kinetics to the non-phosphorylated Cc but presented a −45 mV midpoint redox potential shift. This mutation also inhibits the activation of caspase downstream signaling in vitro[39]. Nevertheless, the Y48E mutation leads to *ca.* 20% decrease in the volume of residue Tyr48, whereas it is increased upon phosphorylation. Thus, using Y48E to mimic phosphorylated Y48 underrepresents volume-related effects[40]. Therefore, the incorporation of a noncanonical amino acid that presents volume and charge values closer to those of phosphotyrosine is a better alternative. Among the different noncanonical amino acids that can be used to mimic Tyr-phosphorylation, *p*-carboxy-methyl-L-phenylalanine (*p*CMF) is one of the best substitutes, as it is stable, efficiently incorporated by *E. coli* cells, and preserves a similar volume than the phosphotyrosine residue[41]. This mutation does not affect the overall structure and heme iron environment as Tyr48-phosphorylated Cc (Supplementary Fig. 1)[42]. In terms of functionality, the Y48*p*CMF Cc shows lower binding affinity for the *distal* sites of $Cc_1$ and CcO, altering the diffusion pathway of Cc molecules through $Cbc_1$-CcO, thus modulating the ETC flux[42]. Finally, the Y48*p*CMF mutation effectively enhances peroxidase activity and induces an antiapoptotic function of Cc[42].

Our recent observation of inter-protein long-distance ET between $Cc_1$ and Cc through the aqueous solution[4] can be explained by the presence of a charge conduit at the active interface between the two proteins (Gouy-Chapman conduit). These results suggest that ET could occur already when the two proteins are approaching (i.e., in the encounter state) without the need of establishing a well-defined, static protein complex, thereby reconciling high specificity with weak binding, and allowing to keep the high turnover rate required in the ETC[5]. Within this scenario, the presence of a charge conduit is of paramount importance for a sustained electron transport. The spatial extent of the charge conduit was found to depend on the redox potential of the interacting proteins and on the charge distribution between them.

Here we aim to investigate the molecular mechanism of ET regulation by phosphorylation in Cc-$Cc_1$. We deploy an array of single-molecule, bulk, and computational methods to address this question in wildtype, phosphomimetic, and phosphorylated Cc. Our findings show that phosphorylation strengthens the interaction with $Cc_1$ (lower equilibrium dissociation constant−$K_D$−) and disrupts the Gouy-Chapman conduit associated with long-distance charge transport. These results could be of general significance for inter-protein ET, its regulation under diverse conditions in the cell, and the switching between multiple tasks.

## Results
### Long-distance ET between Cc-$Cc_1$ partners is impaired by Cc Tyr48 phosphorylation
The effects of Tyr48 replacement by the phosphomimetic *p*CMF in the inter-protein ET between Cc and the mitochondrial complex III, or C$bc_1$, were studied using the cross complex between human Cc and the soluble domain of plant cytochrome $Cc_1$—which contains all residues of the physiological mature (without mitochondrial transit peptide) protein until Glu265, at which the C-terminal hydrophobic helix starts[12]—by means of electrochemical scanning tunneling microscopy (EC-STM)[4]. Cc and $Cc_1$ were immobilized on the probe and the sample electrodes of EC-STM, respectively, with their redox active sites facing the solution, thereby allowing protein interaction. $Cc_1$ was immobilized through its native N-terminal Cys10 to the Au(111) sample electrode of the EC-STM, and Cc or Y48*p*CMF Cc were anchored to the gold

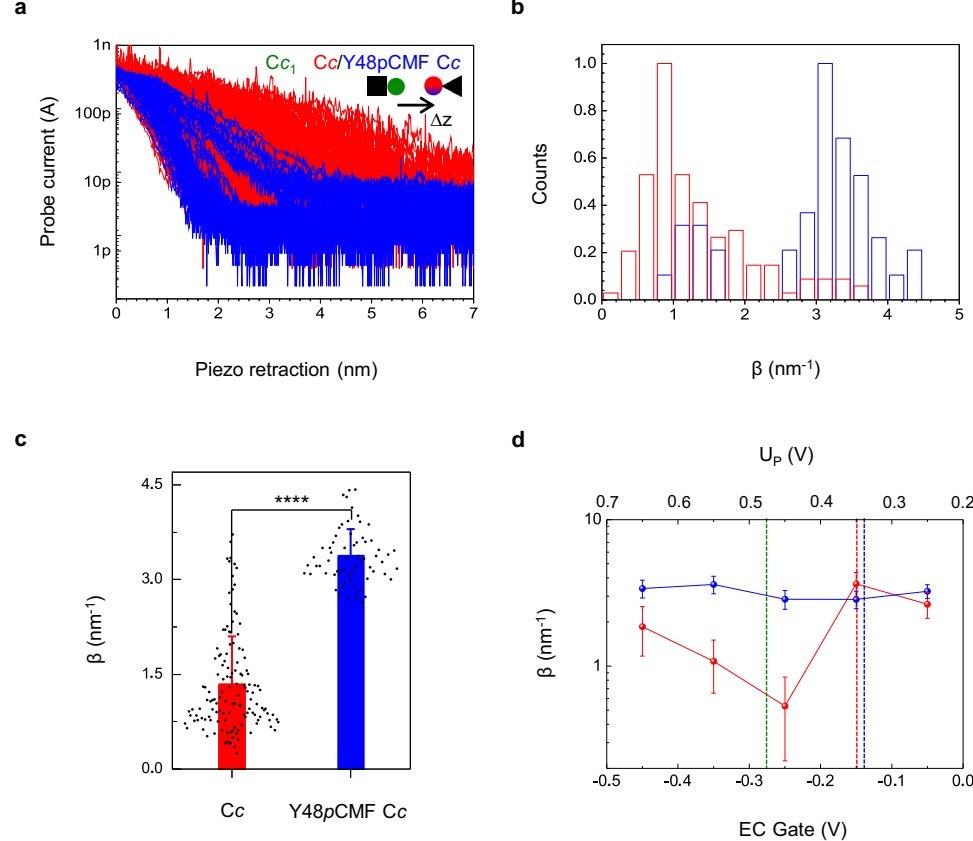

**Fig. 1 | Current-distance electrochemical tunneling spectroscopy of $Cc_1$-$Cc$.**
**a** Ensemble of semi-logarithmic current-distance (I-z) curves during probe retraction, showing a more abrupt current decay for $Cc_1$-Y48pCMF $Cc$. Sample and probe electrodes are represented by a square and a triangle, respectively. Us = −200 mV vs. SSC and at constant bias of 800 mV. **b** Histograms of distance decay factors (β) quantified from individual curves in **a**. **c** plot of the averaged β values (mean ± s.d.) β = 1.3 ± 0.8 nm⁻¹ for $Cc_1$-$Cc$ (n = 103 I-z curves from 2 independent experiments, red) and β = 3.1 ± 0.4 nm⁻¹ $Cc_1$-Y48pCMF $Cc$ (n = 64 I-z curves from 2 independent experiments, blue), respectively, showing significant differences (two-sample t-test, t Statistic = 19.65, degrees of freedom = 195, **** $P < 0.0001$). **d** Averaged β

values (mean ± s.d. of n = 80) vs. EC gate potential at 200 mV constant bias obtained for $Cc_1$-$Cc$ (red) and $Cc_1$-Y48pCMF $Cc$ (blue). The midpoint redox potentials for $Cc_1$ at the sample (0.276 V; green dashed line) and $Cc$ or Y48pCMF $Cc$ (red and blue dashed lines, respectively) at the probe are indicated, showing the β minimum for $Cc_1$-$Cc$ is located within the midpoint redox potentials of $Cc_1$ and $Cc$. Despite being redox-active at similar potential than $Cc$, Y48pCMF $Cc$ did not display a β minimum within the measurement range. All experiments were performed in 50 mM phosphate buffer, pH 6.5. Initial current set point 0.4 nA. Averaged β values for $Cc_1$-$Cc$ (red) are reproduced from reference 4[4]. Source data are provided as a Source Data file.

probe of the EC-STM through the C-terminal mutation E104C, thus facilitating the homogenous orientation of the proteins, while still preserving rotational freedom and good conductivity[43].

The experiments were conducted as described[4] in an electrochemical cell under bipotentiostatic control of the probe and sample electrodes versus an Ag/AgCl (SSC) reference electrode, spanning a potential range around the redox potentials of $Cc_1$ and $Cc$, and always keeping $Cc_1$ reduced (e.g. sample potential ($U_S$) = −0.20 V) and $Cc$ oxidized (e.g. probe potential ($U_P$) = 0.60 V). These values yield a constant positive bias $U_{bias} = U_P - U_S$ = 0.80 V by which the electrons are transferred from $Cc_1$ to $Cc$, in analogy to physiological ET.

Current-distance (I-z) electrochemical spectroscopy was conducted between $Cc_1$-$Cc$ and $Cc_1$-Y48pCMF $Cc$ in 50 mM phosphate buffer pH 6.5. By positioning the EC-STM probe over the sample at a current set point of 0.4 nA, and after a stabilization period, the feedback loop was briefly disconnected, and the probe current recorded while the piezoelectric scanner holding the probe was retracted 15 nm from the set point at 12 nm·s⁻¹. Then, the feedback was restored, and the probe reengaged, allowing subsequent recordings at different areas of the sample. Faradaic leakage current was maintained below a few pA over the entire recording range through probe insulation[44]. A nearly exponential current decay was observed in both cases from the initial current set point up to the distance-independent faradaic

leakage current. For $Cc_1$-$Cc$ the current decay spans several nanometers as previously reported[4], while for $Cc_1$-Y48pCMF $Cc$ the current decays more abruptly, reaching leakage values 3 nm away from the set point (Fig. 1a). Fitting the exponential region of the curves yields the distance decay factor (β) distributions shown in Fig. 1b, which are centered at β = 1.3 ± 0.8 nm⁻¹ for $Cc_1$-$Cc$, and at $β_1$ = 1.3 ± 0.2 nm⁻¹ and $β_2$ = 3.1 ± 0.4 nm⁻¹ for $Cc_1$-Y48pCMF $Cc$ showing a bimodal distribution (Fig. 1c). Although $β_1$ values indicate that long-distance ET still takes place for a small population of recordings (up to 22%) in $Cc_1$-Y48pCMF $Cc$, $β_2$ population doubling the β obtained for $Cc$ suggests that the replacement of Tyr48 by the phosphomimetic pCMF causes a severe alteration of the inter-protein ET. The presence of the phosphomimetic group can affect inter-protein ET by perturbing the two main steps involved in the process. Namely, modifying intra-protein conduction pathways at the protein level or altering the interface between the two proteins and thus affecting the charge conduit. When we compared the β values between bare gold and either $Cc$ or Y48pCMF $Cc$, we found no significant differences (Supplementary Fig. 2), thus indicating that the Y48pCMF mutation does not modify protein conduction, and that the changes observed in Fig. 1a, b may involve the charge conduit that is established between $Cc_1$ and $Cc$.

Electrochemical gating experiments were conducted for the $Cc_1$-Y48pCMF $Cc$ interaction. We recorded I-z plots at different probe and

sample potentials keeping the bias constant. The potential of the reference electrode was adjusted to modulate the gate voltage in analogy to the gate electrode in a field effect transistor. As the EC-STM probe is grounded in our STM electronic configuration, the EC gate corresponds to -$U_S$ at any given bias[45]. For $Cc_1$-$Cc$ we have previously observed that $\beta$ displays a marked potential dependence at a moderate bias of 200 mV. Averaged $\beta$ values showed a minimum ($\beta = 0.5 \pm 0.3$ nm$^{-1}$) at an EC gate potential of $-0.25$ V, which was located between the midpoint redox potentials of $Cc_1$ (0.28 V vs. SSC) and $Cc$ (0.35 V vs. SSC), respectively (Fig. 1d, red curve). This potential-dependent effect allowed to measure current spanning more than 10 nm away from the departure set point and highlights the contribution of the redox sites in the ET between $Cc_1$ and $Cc$[4]. The $\beta$ values of $Cc_1$-Y48$p$CMF $Cc$ do not depend on the EC gate within the considered redox potential window (Fig. 1d, blue curve), in striking contrast to $Cc_1$-$Cc$. The midpoint redox potential of immobilized Y48$p$CMF $Cc$ (0.36 V vs. SSC), is very similar to that of $Cc$ (Supplementary Fig. 2), as previously reported[41], and contrary to the Y48E substitution, which shows a ~ 80 mV reduction of its midpoint redox potential with respect to $Cc$[39]. The redox potential values obtained are within the previously reported values for electron-transfer active $Cc$ directly immobilized on an Au electrode[46,47]. Therefore, Y48$p$CMF $Cc$ maintains its redox activity but cannot be engaged in long-distance ET through the aqueous solution. This is likely due to an alteration in the charge conduit between the protein partners rather than coarse protein structural changes. $Cc$ structure consists of five cooperative folding-unfolding units (foldons)[48] and its midpoint redox potential is very sensitive to protein folding and heme group coordination[49,50]. As Met80 is located in the second most unstable foldon (foldon IV) of $Cc$, subtle alterations of $Cc$ structure affect the Met80-Fe coordination bond and, therefore, the redox potential value. Indeed, several authors have demonstrated the important role of Met80-Fe coordination on $Cc$ redox potential, as the loss of this bond shifts the redox potential towards negative values ($-200$ mV)[51]. Based on our voltammograms (Supplementary Fig. 2), we can rule out the hypothesis of partial or substantial denaturation of the protein after its immobilization on the Au(111) surface, since no voltammetric signal is observed at negative redox potential values.

To complement current-distance spectroscopy measurements, we studied current blinks in the static break-junction approach[52,53] to measure the spontaneous interaction between protein partners (Fig. 2). By setting the current set point to 0.30 – 0.35 nA, and transiently disconnecting the feedback loop at a constant bias of 0.8 V, we recorded current vs. time ($I$-$t$) traces on $Cc_1$-$Cc$ and $Cc_1$-Y48$p$CMF $Cc$ interaction (Fig. 2a, red and blue, respectively). In the $I$-$t$ curves, sudden jumps appear in the measured current (blinks), which are attributed to the spontaneous contact between the probe and the sample. The collected blink currents ($I_{blink}$), after subtracting the baseline current ($I_{baseline}$) were set to a common time origin (initial blinking time = 0) and plotted with the blinking lifetime in 2D-histograms (Fig. 2b). These 2D-blinking maps show the conductance dispersion of the partners' interaction. Taking the most probable conductance peak, we found that $Cc_1$-Y48$p$CMF $Cc$ interaction led to a significantly higher conductance (P < 0.05) (Fig. 2c). Gaussian fitting of main conductance peaks yields $G = (4.3 \cdot 10^{-6} \pm 9 \cdot 10^{-7}) \cdot G_0$ for $Cc_1$-$Cc$ and $G = (7.4 \cdot 10^{-6} \pm 5 \cdot 10^{-7}) \cdot G_0$ for $Cc_1$-Y48$p$CMF $Cc$ (with $G_0 = 2e^2/h$). In Fig. 2d, $I_{blink}$ is plotted against its corresponding $I_{baseline}$ showing that most probable $Cc_1$-$Cc$ interaction ($G = 4.3 \cdot 10^{-6} \pm 9 \cdot 10^{-7}$)·$G_0$ can occur below the set point current (dashed line), thereby at longer distances than with Y48$p$CMF $Cc$, in agreement with $I$-$z$ measurements.

**$Cc$ Tyr48 phosphorylation strengthens the $Cc$-$Cc_1$ interaction**
Since EC-STM results suggest that $Cc$ phosphorylation alters the interface between the two proteins, we further evaluated the interprotein interaction using bulk and single molecule measurements.

We assessed the effect of Tyr48 replacement by the phosphomimetic $p$CMF on the interaction strength between $Cc$ and $Cc_1$ by analyzing the forced unbinding between human $Cc$ and the soluble domain of plant cytochrome $Cc_1$ using single molecule force spectroscopy with an atomic force microscope (AFM)[54]. In addition, binding affinities of the $Cc_1$-$Cc$ complex were calculated from solution-based surface plasmon resonance (SPR) ensemble measurements, and the impact on $Cbc_1$ complex (complex III) activity was assessed in mammalian mitochondrial extracts.

To carry out single-molecule unbinding experiments, $Cc_1$ was immobilized on Au(111) substrates in a similar way as described for the EC-STM experiments. Either $Cc$ or Y48$p$CMF $Cc$, both with the C-terminal mutation E104C, were conjugated to the maleimide group of a PEG$_{27}$ heterobifunctional crosslinker previously anchored to amino-functionalized AFM silicon probes. In the force spectroscopy measurements, the tip and substrate are approached and forced into contact up to a maximum force of 200 pN and then retracted away from the surface at constant velocity, in 50 mM phosphate buffer pH 6.5. The force-separation retraction traces at 1 μm·s$^{-1}$ velocity occasionally displayed unbinding events like those in Fig. 3a, which allows quantifying the force ($F_r$) and length ($l_r$) (see representative example in Fig. 3a inset) at which the contact is ruptured. The fact that the rupture length is maintained around 10–20 nm (Fig. 3b) indicates that protein unfolding is not significant, considering that the linker length is about 8 nm (monomer length of approx. 0.3 nm)[55,56], and the measured distances may be due to deformation and elongation of the proteins prior to dissociation. No correlation is found between $F_r$ and $l_r$ for neither $Cc_1$-$Cc$ nor $Cc_1$-Y48$p$CMF $Cc$ (Fig. 3b). For the pair $Cc_1$-$Cc$, $F_r$ is centered at $55 \pm 17$ pN (fitting a gaussian distribution) with a mean value of 75 pN (Fig. 3b, c). These relatively low rupture forces support the idea of the transient nature of the $Cc_1$-$Cc$ interaction. In contrast, $Cc_1$-Y48$p$CMF $Cc$ display higher unbinding forces (P < 0.05), centered at $77 \pm 24$ pN (gaussian fit) with a mean value of 91 pN (Fig. 3b, c).

We complemented these results with force-mediated dissociation kinetics studies. The lifetime of a bond sustained by weak noncovalent interactions decreases when subjected to force, which is conceptualized as force causing a tilt on the energy landscape, lowering energy barriers[57]. The unbinding force generally depends on the loading rate, as theory predicts for many biological interactions like receptor-ligands bonds[58]. In that case, the measured unbinding force corresponds to an individual point in a continuous spectrum of bond strengths. Therefore, we measured the rupture force at varying retraction velocities (from 0.5 to 4 μm·s$^{-1}$). To avoid transmission of the load through the polymer linker and bring high inaccuracy to the loading rate values, either $Cc$ or Y48$p$CMF $Cc$ were immobilized directly onto Au-coated probes through the thiol-Au chemistry. Here, the protein immobilization and orientation are comparable to those of the ECSTM experiments. The two proteins are then forced into contact and $F_r$ and $l_r$ are evaluated at different retraction velocities. These measurements only carry direct information of the kinetics of unbinding (not binding).

The force of $Cc_1$-$Cc$ unbinding is constant for the pulling velocities studied (contour maps $F_r$ vs. $l_r$, Fig. 3d). In contrast, $Cc_1$-Y48$p$CMF $Cc$ unbinding becomes force resistant, and the unbinding force scales logarithmically with the pulling velocity and the loading rate (Fig. 3d). $Cc_1$-$Cc$ dissociation occurs at low forces independently of the pulling velocity (mean $F_r$ vs velocity in log scale, Fig. 3e). This behavior is associated with two bodies being pulled apart during a near-equilibrium mode[59–61], in which complexes may have short lifetime and unbinding-rebinding is faster than pulling. This allows quick dissociation of the complex at biologically accessible loading rates and forces, in accordance with the requirement of a rapid turnover. In contrast, the rupture force between $Cc_1$-Y48$p$CMF $Cc$ depends logarithmically on the loading rate ($r$) (Fig. 3f), meaning that the measurements are done in a

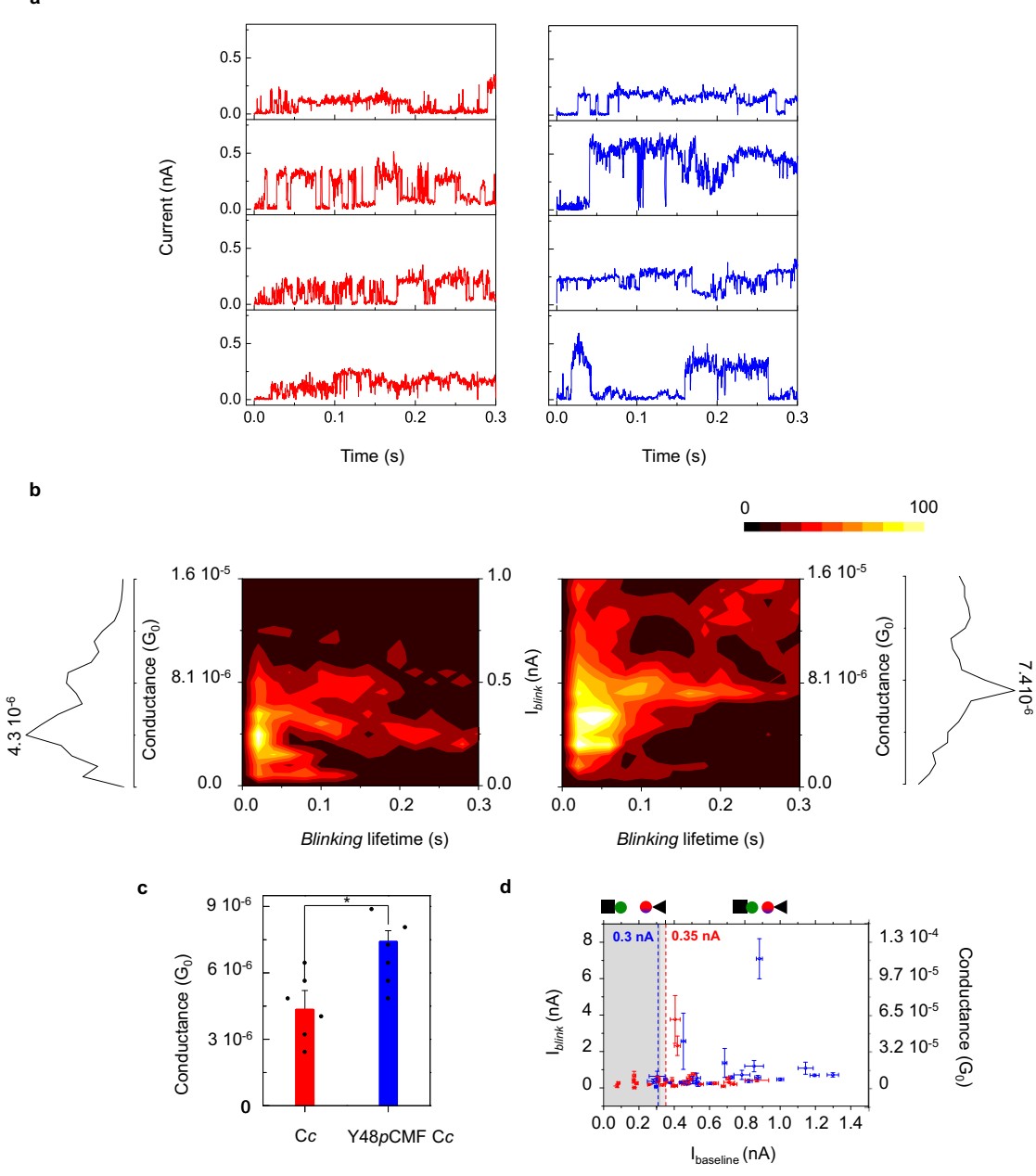

**Fig. 2 | Blinking experiments. a** Representative blinking recordings for $Cc_1$-$Cc$ (red) and $Cc_1$-Y48$p$CMF $Cc$ (blue). Blinks are shown as transient jumps of the current flowing between the two electrodes at a constant distance (imposed by initial current set point) and bias (800 mV). **b** Two-dimensional histograms of blink current ($I_{blink}$) and blink lifetime for $Cc_1$-$Cc$ (left, n = 100980 conductance values included) and for $Cc_1$-Y48$p$CMF $Cc$ (right, n = 252193 conductance values included). Counts have been normalized to a color scale, with 100 counts representing the maximum, and 0 representing the minimum. The corresponding conductance (G) profiles are shown. Baseline correction was applied in a and b. G = $I_{blink}/V_{bias}$ is used to obtain the conductance values. $G_o = 2e^2/h = 77.5$ μS, with $h$ the Planck constant and $e$ the elementary charge. **c** Plot of the averaged G values from

gaussian fit of the most probable conductance peaks (mean ± s.d.)
G = $4.3 \cdot 10^{-6} \pm 9 \cdot 10^{-7}$ $G_o$ for $Cc_1$-$Cc$ (red) and G = $7.4 \cdot 10^{-6} \pm 5 \cdot 10^{-7}$ $G_o$ for $Cc_1$-Y48$p$CMF $Cc$ (blue), respectively, showing significant differences (n = 6 conductance values from 2 independent experiments, two sample t-test, t Statistic = 2.777, degrees of freedom = 10.00, *P = 0.0195. **d** Plot of $I_{blink}$ against its baseline current ($I_{baseline}$) for a set of blinks of $Cc_1$-$Cc$ (red) and $Cc_1$-Y48$p$CMF $Cc$ (blue) (mean ± s.d.). The dashed vertical lines indicate the corresponding current set points at which feedback was turned off. Sample and probe electrodes are represented by a square and a triangle, respectively. All experiments were performed in 50 mM phosphate buffer, pH 6.5. Source data are provided as a Source Data file.

kinetic regime (non-equilibrium bond rupture), as predicted by the Bell-Evans model based on a single transition state theory[57,59,60,62]. The most probable unbinding force plotted against the loading rate on a log scale results in a straight line. Kinetic parameters can be obtained assuming that the energy barrier observed in dynamic force spectroscopy (DFS) remains rate limiting below the loading rate range tested. Following the Bell-Evans model[57,59,62], the rate of the molecular complex force-activated unbinding ($k^o_u$) and the

distance from the bound state to the barrier that is rate limitng for unbinding ($x_u$) can be calculated from a linear fit to the data only for the $Cc_1$-Y48$p$CMF $Cc$ pair ($k^o_u = 48$ s;$^{-1}$ $x_u = 0.31$ nm). Overall, the DFS results indicate that the Y48$p$CMF mutation in $Cc$ strengthens the interaction with $Cc_1$, thereby hindering a fast turnover.

$Cc_1$-$Cc$ binding affinities were calculated from solution-based surface plasmon resonance (SPR) ensemble measurements. Protein binding affinities are commonly measured and described by the

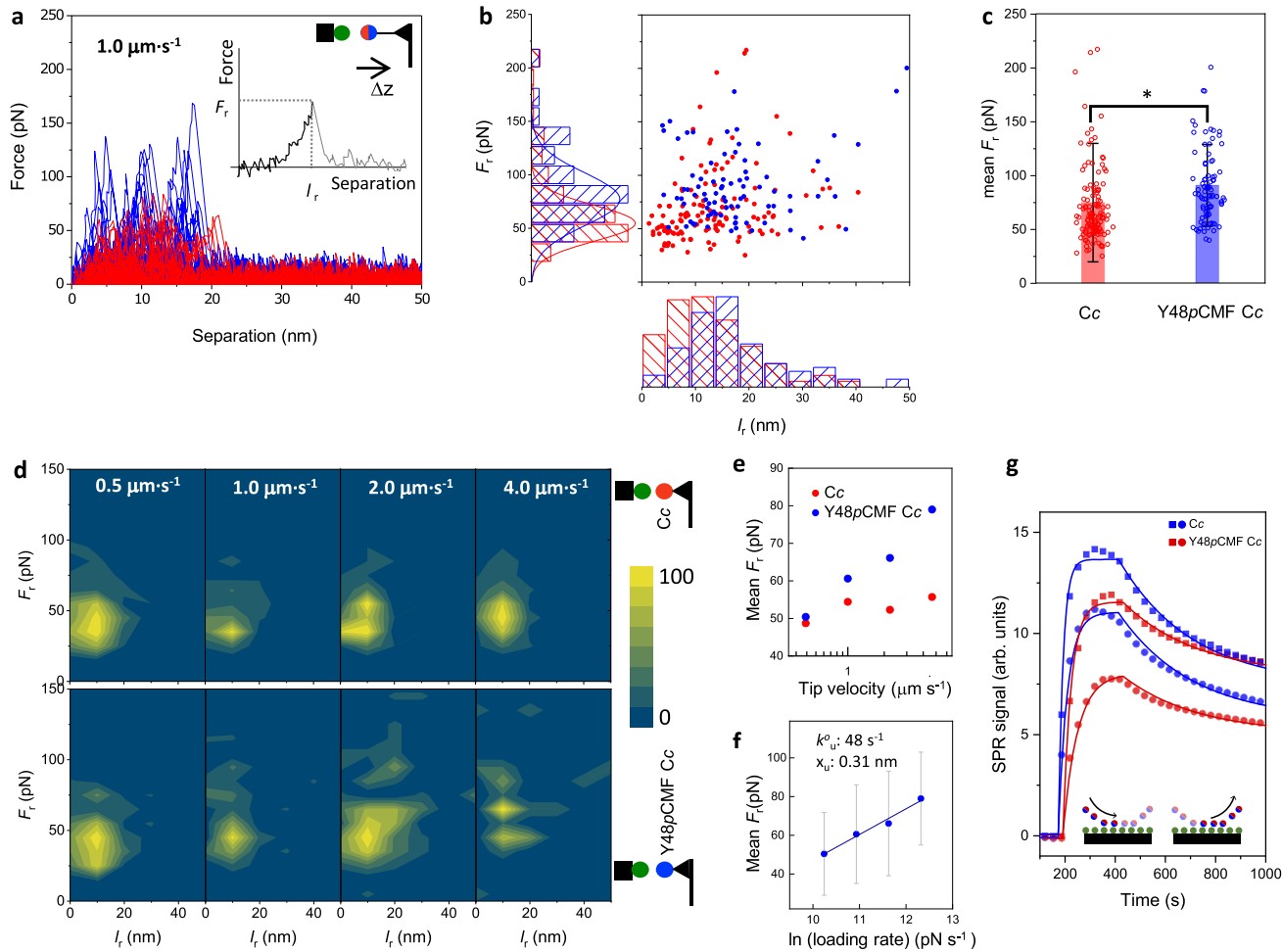

**Fig. 3 | Unbinding between $Cc_1$-$Cc$ studied by AFM-FS single molecule and surface plasmon resonance (SPR) ensemble measurements. a** Representative force-separation unbinding events upon tip retraction at 1 μm·s⁻¹ for $Cc_1$-$Cc$ (red) and $Cc_1$-Y48$p$CMF $Cc$ (blue), when $Cc_1$ is directly anchored to Au(111) and $Cc$ is anchored to the AFM tip through a PEG$_{27}$ linker. The inset shows the parameters $F_r$ and $l_r$, that are collected from each curve and displayed in **b** as $F_r$ $vs.$ $l_r$ with the corresponding histograms in each axis. $F_r$ distributions are fitted to a Gaussian. **c** Plot of the averaged $F_r$ values (mean ± s.d.) for $Cc_1$-$Cc$ (n = 134 force-separation curves from 2 independent experiments, red) and $Cc_1$-Y48$p$CMF $Cc$ (n = 88 force-separation curves from 3 independent experiments, blue), showing significant differences (two sample t-test, t Statistic = 2.34, degrees of freedom = 220, *P = 0.02). **d** $F_r$ $vs.$ $l_r$ contour maps for the unbinding at different pulling rates (0.5 − 4.0 μm·s⁻¹) for $Cc_1$-$Cc$ (top, n = 212, 185, 55 and 226 for 0.5, 1.0, 2.0 and 4.0 μm·s⁻¹,

respectively) and $Cc_1$-Y48$p$CMF $Cc$ (bottom, n = 194, 83, 72 and 97 for 0.5, 1.0, 2.0 and 4.0 μm·s⁻¹, respectively), when $Cc_1$ is directly anchored to Au(111) and $Cc$ is directly anchored to the Au-coated AFM tip. **e** Mean $F_r$ vs. tip velocity (in log scale). **f** Average $F_r$ (mean ± s.d.) $vs.$ ln (loading rate) for the $Cc_1$-Y48$p$CMF $Cc$ unbinding fitted to a linear function (blue line) (n = 194, 83, 72 and 97, for 28000, 56000, 112000 and 224000 pN·s⁻¹, respectively). Forced-activated unbinding parameters are calculated following the Bell-Evans model[57,59,62]. All AFM experiments were performed in 50 mM PBS, pH 6.5. **g** Binding kinetics determined from SPR measurements. SPR sensograms obtained with 5 μM (squares) and 2.5 μM (circles) of $Cc$ (red) and Y48$p$CMF $Cc$ (blue) analyte and immobilized $Cc_1$. Lines correspond to a simultaneous, global fit of the two curves to a one-site binin model. Data is representative of a total of 16 spots. The binding assays were performed in 10 mM PBS, pH 6.5. Source data are provided as a Source Data file.

equilibrium dissociation constant ($K_D$), which evaluates the strength of biomolecular interactions (i.e., the greater the $K_D$ value, the weaker the attraction between both protein partners). Fitting of the SPR curves for the interaction between $Cc$ (WT and Y48$p$CMF) and $Cc_1$ to a 1:1 binding model (Fig. 3g) allows to obtain the association ($k_{on}$) and dissociation ($k_{off}$) rates for the interaction (Table 1). The calculated association rate $k_{on}$ of $Cc_1$-Y48$p$CMF $Cc$ complex is nearly 2-fold larger than that of the $Cc_1$-$Cc$ complex, whereas the differences between spontaneous off-rate $k_{off}$ values are small (but statistically significant, P < 0.05). Differences found between $k^o_u$ values determined in single-molecule DFS experiments and spontaneous off-rates $k_{off}$ values determined by SPR have been attributed to the complexity of the energy landscape that may cause high differences between the forced unbinding mechanism and the thermally activated one[63,64].

The differences in $K_D$ ($k_{off}/k_{on}$) values between $Cc$ and Y48$p$CMF $Cc$ in complex with $Cc_1$ go along with the single molecule

force spectroscopy results, and agree with published NMR and ITC measurements[42], demonstrating a stronger interaction force between $Cc_1$-$Cc$ when the Tyr48 residue is replaced by $p$CMF.

We further investigated if the observed differences in the unbinding forces and $K_D$ values could modulate the mitochondrial complex III activity (Supplementary Fig. 4). The biochemical assay to study the complex III activity relies on following the changes in $Cc$ redox state from oxidized to reduced, by recording the absorption spectrum of $Cc$ at 550 nm. Our results with mammalian mitochondrial extracts showed that Y48$p$CMF $Cc$ displays a 1.6-fold reduction (P < 0.001) of the net complex III activity (0.28 nmol·min⁻¹·μg⁻¹ $vs.$ 0.17 nmol·min⁻¹·μg⁻¹ for WT and Y48$p$CMF $Cc$, respectively). Such results support the hypothesis that the observed differences in unbinding forces and $K_D$ values could be physiologically relevant in the modulation of the ET behavior of Tyr48-phosphorylated $Cc$.

**Table 1 | Analysis of surface plasmon resonance (SPR) curves**

| Spot | Analyte | $k_{on}$ (M$^{-1}$ s$^{-1}$) | $k_{off}$ (s$^{-1}$) | $K_D$ (M) |
|------|---------|------------------------------|----------------------|-----------|
| Cc$_1$ | Cc | $6.54{\cdot}10^3 \pm 8.8{\cdot}10^1$ | $3.18{\cdot}10^{-3} \pm 5{\cdot}10^{-5}$ | $5.27{\cdot}10^{-7} \pm 1.4{\cdot}10^{-8}$ |
| | Y48$p$CMF Cc | $1.00{\cdot}10^4 \pm 1.57{\cdot}10^2$ | $3.58{\cdot}10^{-3} \pm 6{\cdot}10^{-5}$ | $3.57{\cdot}10^{-7} \pm 9{\cdot}10^{-9}$ |

Association rate ($k_{on}$) and spontaneous off-rate ($k_{off}$) constants and dissociation equilibrium constants ($K_D$) were calculated from the fits of SPR curves to 1:1 binding model. The errors reported are the standard errors derived from the fitting of the experimental data to a 1:1 binding model. $R^2$ are 0.9975 and 0.9954 for WT and Y48$p$CMF Cc fitting, respectively.

## Cc phosphorylation disrupts the Gouy-Chapman conduit for long-distance ET with Cc$_1$

Long-distance currents between Cc and Cc$_1$ can be explained by a low ionic concentration in the aqueous gap between the two protein redox sites, compared to the solution bulk[4]. Cation depletion near the redox-active sites impairs charge screening in the solution and allows the electric field to extend several nanometers within the diffuse ionic Gouy-Chapman layer. To account for the changes in β observed when Tyr48 is replaced by the phosphomimetic $p$CMF in the EC-STM experiments, we performed molecular dynamics (MD) calculations of the pairs Cc$_1$-Cc (WT) and Cc$_1$-Y48$p$CMF Cc (phosphomimetic), as well as Cc$_1$-pY48 Cc (phosphorylated Tyr48: O-phospho-L-tyrosine) with charge −1 and −2. Note that the latter "true" phosphorylated Cc forms offer valuable insights as they are not yet experimentally accessible. The proteins were separated 3.2 nm with the redox active sites facing each other, in a solvent with explicit water molecules and 50 mM NaCl concentration (Supplementary Fig. 3a–d)

Cation depletion near the redox-active sites in the aqueous gap between the proteins is observed for Cc$_1$-Cc, and the concentration is nearly zero in the proximity of the redox active sites (Fig. 4a), in accordance with our previous studies[4]. In contrast, for Cc$_1$-Y48$p$CMF Cc an accumulation of cations is found around the proteins and in the region between them (Fig. 4b, h), likely due to the extra negative charge at position 48 that alters the surface electrostatic potential. This is reflected in the computed electrostatic potential in a 50 mM ionic concentration, which shows equipotential lines that extend between Cc and Cc$_1$ proteins, forming a conduit (Fig. 4c), but this is reduced between Y48$p$CMF Cc and Cc$_1$ (Fig. 4d). For Cc$_1$-Cc the space between both heme groups had the lowest average concentration of sodium ions (0.018 M) in comparison to Cc$_1$-Y48$p$CMF Cc (0.067 M). Thus, MD calculations show that the Y48$p$CMF mutation of Cc causes the disruption of the Gouy-Chapman conduit that has been associated to long-distance ET through the aqueous solution in agreement with EC-STM results. Interestingly, the phosphorylation of Tyr48 in native Cc may shift the electrostatic field even more than the Y48$p$CMF mutation, as evidenced when the computed average sodium concentration (average sodium concentration of 0.044 M for charge −1 and 0.052 M for charge −2, Fig. 4h, Supplementary Fig. 3e, f) and electrostatic potentials are obtained for pY48 Cc with charge −1 and −2 (Figs. 4e, f, h). In both cases, the equipotential lines do not extend between Cc and Cc$_1$.

We also performed Brownian dynamics (BD) simulations to calculate binding kinetics using the MD structures of Cc and Y48$p$CMF Cc complexes with Cc$_1$. For each computation, two curves are represented (Fig. 4i), corresponding to 4 independent pair-distances criteria for reaction, as reported previously[65]. We estimated association rate ($k_{on}$) values at a distance of 0.6 nm as reaction criterion (vertical line in Fig. 4i). Cc and Y48$p$CMF Cc species show different behaviors across the simulation volume (expressed as distance cut-off dependencies of observables): i) the long-range attraction between Y48$p$CMF Cc and Cc$_1$ is larger than between Cc and Cc$_1$, and ii) below 12 nm, the estimated apparent $k_{on}$ values for Y48$p$CMF Cc decay more steeply than in the case of Cc, probably due to bias of electrostatic steering. This matches the differences in force values found in DFS experiments,

indicating that the Y48$p$CMF mutation strengthens the interaction with Cc$_1$, as pointed out before. The ratio between the $k_{on}$ values decreases hyperbolically with the distance, reflecting the impact of changes in the energetic landscape of interaction. The rate constant $k_{trigger}$ was set at the cut-off distance (2.5 nm) at which ET can be triggered in the EC-STM experiments. At the contact level, reckoned $k_{on}$ for Cc$_1$-Y48$p$CMF Cc are ca. 10-fold larger than those estimated for Cc$_1$-Cc (Table 2, Fig. 4i). At the cut-off distance of 2.5 nm, $k_{trigger}$ values increase 2-3-fold in the trajectories corresponding to Cc$_1$-Y48$p$CMF Cc compared to Cc$_1$-Cc. In summary, Cc$_1$ attracts Y48$p$CMF Cc stronger than the WT Cc, even at long distances.

## Discussion

Phosphorylation is one of the most prevalent mechanisms of regulation of protein activity. In particular, tyrosine phosphorylation plays a crucial role in cellular processes such as metabolism or cellular growth. Cc and its partner CcO are among the mitochondrial proteins whose function is highly regulated by phosphorylation. Cc phosphorylation has been suggested to perform a crucial role in regulating mitochondrial respiration and cell fate[30].

Tyr48 phosphorylation has been reported to affect the binding affinity of Cc towards its molecular partners, altering the diffusion pathway of Cc molecules through Cbc$_1$-CcO and affecting the ETC flux by partially inhibiting the mitochondrial respiration[42]. However, the molecular mechanism through which Cc phosphorylation impacts the ETC is unknown. The multimodal experiments on phosphomimetic Cc and calculations reported here show that upon phosphorylation the long-distance charge conduit established between Cc-Cc$_1$ is shut off and their interaction is strengthened and departed from binding/unbinding equilibrium in the measured conditions. Overall, these results are in full agreement with the consensus evidence published on the topic and unveil a nanoscopic view and mechanistic framework that offers deeper insights into the interaction between redox protein partners of the respiratory ETC.

Notably, these results indicate a regulatory advantage of using electrochemically gated, long-distance electron transport through the solution between redox partner proteins. Besides reconciling high specificity and electron transport efficacy with weak binding to keep high turnover rates[4], the Gouy-Chapman conduit bearing reduced ionic density in the confined volume between the proteins can be externally controlled to reduce charge flow. When phosphorylated at Tyr48 (herein mimicked by replacement with $p$CMF), Cc sliding from Cbc$_1$ to CcO is impaired[42]. The stronger interaction between the partners upon phosphorylation was considered to hinder the Cc turnover (exchange with the partner) as it would lead to a frozen complex as concluded from calorimetry data and NMR spectroscopy results[42]. Note that our scanning probe microscopies experiments allow probing interactions at the single protein level, and in the case of EC-STM (I-z and blinking studies) it is possible to keep Cc$_1$ reduced, and Cc oxidized to mimic the physiological situation of ET from Cc$_1$ to Cc. Independent control of probe and sample electrochemical potential in the single molecule force spectroscopy AFM setup is more challenging[66,67] and constitutes one aspect to improve experimentally in future studies.

We observe that electrochemically gated long-range ET is impaired in single-molecule experiments with phosphomimetic Y48$p$CMF Cc, with current decay distances reducing to half (to 3 nm or less, Fig. 1a–c) and abolishing EC gate potential control (Fig. 1d). MD simulations show that the negative charge introduced by the $p$CMF residue on the Cc surface prevents cation exclusion between the proteins' redox active sites, thereby enabling charge screening in this confined interface and disrupting the Gouy-Chapman conduit (Fig. 4). The effects are even stronger in simulations with phosphorylated residues, which bear high relevance as they are not yet experimentally accessible. Besides, we observe that the single molecule unbinding of

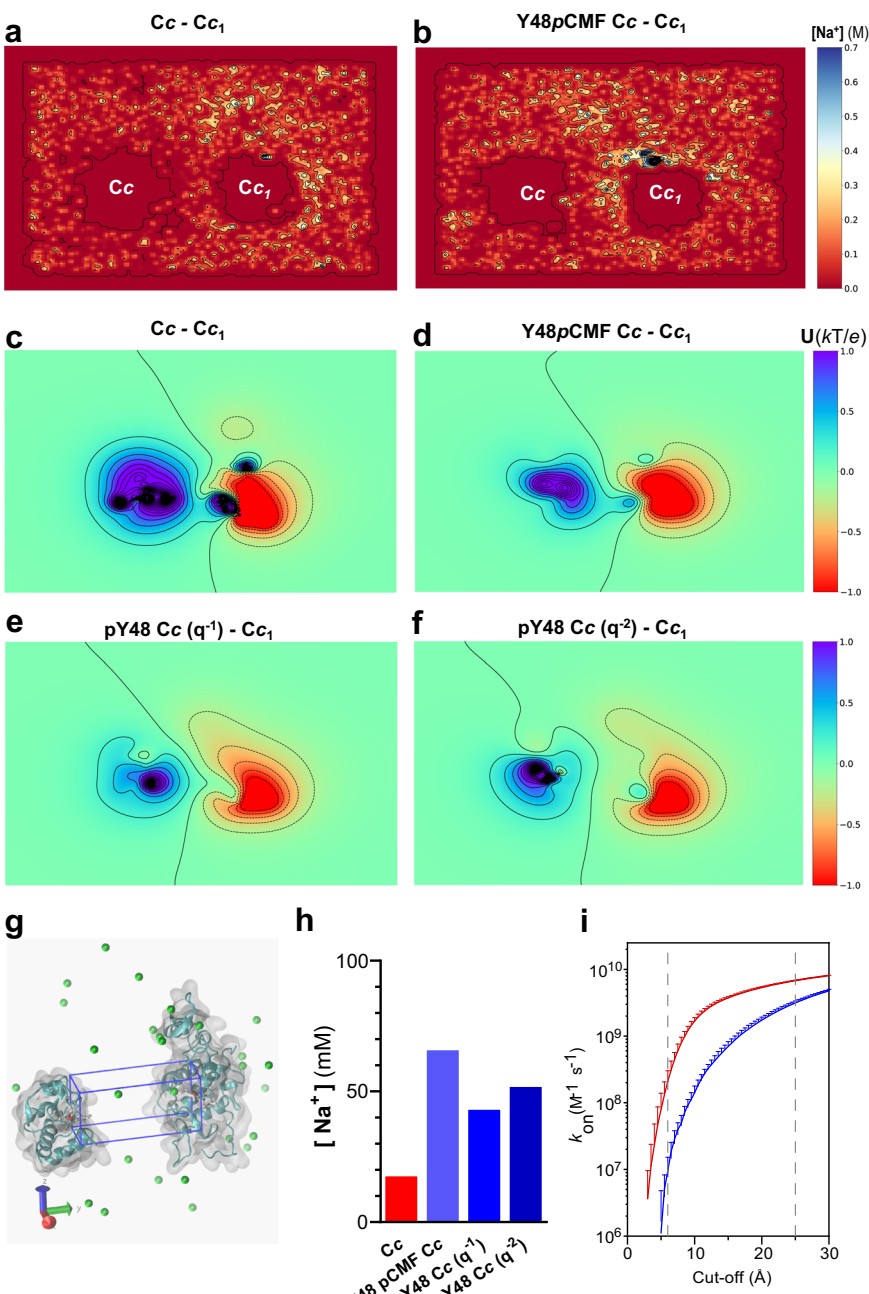

**Fig. 4 | Molecular dynamics (MD) calculations of C$c_1$ and C$c$ display an inter-protein Gouy-Chapman conduit that is disrupted by phosphorylation. a, b** Side view of the averaged sodium concentration map (M) at the X-axis plane dissecting both C$c_1$-C$c$ redox active sites (C$c$ and Y48$p$CMF C$c$, respectively) facing each other at 3.2 nm separation taking the most external hydrogen atoms from the -CH$_3$ distal pairs from each heme group, that is 4.2 nm taking the Fe-to-Fe distance. MD simulations were performed with explicit water (TIP3P solvent model) and 50 mM ionic concentration (NaCl) over 36 ns. Each contour line displays an increase of 0.15 M. (**c–f**) Side view of computed electrostatic potential with APBS[76] from −1 kT/e (red) to 1 kT/e (blue) for C$c_1$-C$c$ pairs: C$c_1$-C$c$ (**c**), C$c_1$-Y48$p$CMF C$c$ in silico (**d**), C$c_1$-pY48 C$c$ (O-phospho-L-tyrosine) with charge −1 (**e**) and C$c_1$-pY48 C$c$ (O-phospho-L-tyrosine) with charge −2 (**f**). Ion concentration is 50 mM, pH 6.5. Each contour line displays an increase of 0.2 kT/e. **g** Virtualization of the C$c_1$-C$c$ system in 3D. Representation of the prism (blue) used to calculate the average concentration of sodium ions throughout the simulation. The image shows a single frame of the C$c_1$-C$c$ system depicting protein (cyan, translucent grey surface), heme groups (white with Iron atom in red), sodium ions (green spheres), and omitting water molecules. The closest backbone residues to the generated prism vertices are His18, Pro76, Lys79 and Ile85 of the Y48$p$CMF C$c$, and Leu134, Leu94, Glu101 and Ala160 of C$c_1$. **h** Average sodium concentration over 36 ns in a 12.9 nm³ volume prism (prism dimensions for x, y, z = 15, 43, 20 units) centered between both heme groups of C$c$ and C$c_1$ for all systems. Volmap Tool Plugin 1.1 from VMD[75] was used. **i** Derivation of binding kinetics from the BD computations of C$c_1$-C$c$ (red) and C$c_1$-Y48$p$CMF C$c$ (blue) complexes. For each computation, two curves are represented, corresponding to 4 independent pair-distances criteria for reaction, as reported previously[65]. Vertical lines correspond to the distances used as reaction criterion to estimate $k_{on}$ and $k_{trigger}$ values. Source data are provided as a Source Data file.

Y48$p$CMF C$c$ from the redox partner C$c_1$ occurs at higher forces (Fig. 3). These results are consistent with the higher binding affinities of phosphomimetic C$c$ compared to the wildtype in bulk experiments (Fig. 3g, Table 1) and BD calculations (Fig. 4i, Table 2).

Figure 5 recapitulates all these results and provides a mechanistic framework showing how phosphorylation of C$c$ alters several functions of this multifaceted protein involved in electron transport and cell fate. First, by shutting down its ability to exchange charge with

**Table 2 | Analysis of brownian dynamics (BD) simulations for the $Cc_1$-$Cc$ interaction**

|  | $Cc_1$-$Cc$ | $Cc_1$-Y48$p$CMF $Cc$ |
|---|---|---|
| $k_{on}$ (M$^{-1}$ s$^{-1}$) (0.6 nm) | $9.50 \cdot 10^6 \pm 7.90 \cdot 10^6$ | $2.10 \cdot 10^8 \pm 9.6 \cdot 10^7$ |
| $k_{trigger}$ (M$^{-1}$ s$^{-1}$) (2.5 nm) | $3.20 \cdot 10^9 \pm 8.6 \cdot 10^7$ | $6.80 \cdot 10^9 \pm 1.4 \cdot 10^8$ |

Estimated association rate at contact level ($k_{on}$) and at the cut-off distance of 2.5 nm ($k_{trigger}$). The errors reported correspond to the standard deviation.

$Cc_1$ (complex III) at long distances through the aqueous solution and in an electrochemically selective manner[4]. And second, by promoting a stronger contact with $Cc_1$ that disturbs the reversible binding found in equilibrium conditions (low forces) and likely reduces the turnover of molecules at complex III. In this scenario, even small amounts of phosphorylated $Cc$ will lead to one of them being stuck at complex III and effectively preventing others from transferring charge. The dissociation constant ($K_D$) is lower for the $Cc$ Y48$p$CMF variant, which must get closer to complex III than the wildtype in order to achieve similar current levels (due to the higher $\beta$ of the former, see Fig. 1a). In this situation, the interaction forces (unbinding force, Fig. 3; binding affinity, Table 1 and Fig. 3g; association rate, Table 2 and Fig. 4i) are higher and thereby the overall turnover of $Cc$ molecules is reduced, lowering the complex III activity (Supplementary Fig. 4). The higher conductance observed in the Y48$p$CMF variant compared to the WT species (Fig. 2c) might be caused by such closer interaction.

In conclusion, we have performed a multimodal study of the interaction between redox partners $Cc$ and $Cc_1$ that covers single protein conductance and force measurements, bulk binding and biochemical assays, and molecular dynamics calculations. By studying a phosphomimetic mutation, we have focused on the impact of phosphorylation in all the observables and integrated the obtained results within the available biochemical evidence, aiming to provide a nanoscopic view and mechanistic framework that was lacking in this topic. Phosphorylation of $Cc$ impairs electrochemically gated long-distance electron transport through the solution and disturbs the binding/unbinding equilibrium between the partners, strengthening their interaction. Calculations show that the cation exclusion volume that is established between the partners (Gouy-Chapman conduit) is disrupted by the negative charges introduced near the $Cc$ redox site by phosphorylation and its experimental molecular mimics. These results allow a deeper understanding of the tradeoff between binding, transfer, and turnover that is established between redox protein partners in the respiratory ETC and provide mechanistic insights into the molecular sites that regulate cell signaling circuits.

## Methods

### Protein expression and purification
*Escherichia coli* (*E. coli*) BL21 (DE3) cells were transformed with pBTR1-WT plasmid—that contained the *CYCS* gene-coding for human $Cc$ and, the *CYC3* gene of the yeast $Cc$ heme lyase, required for the maturation of $Cc$—to recombinantly express *Homo sapiens* WT $Cc$ for Surface Plasmon Resonance (SPR) experiments. For the expression of E104C mutant, used for Electrochemical Scanning Tunneling Microscope (EC-STM) and Atomic Force Microscopy-Force Spectroscopy (AFM-FS) measurements, pBTR-1 plasmid was mutated by replacing the GAA triplet corresponding to Glu104 by TGT, which codifies to cysteine residue. The primers for PCR were pBTR1-E104C fw (5′-AAGGCGACGAACTGTTGATAAGGTACCA-3′) and pBTR1-E104C rv (5′-TGGTACCTTATCAACAGTTCGTCGCCTT-3′). For this purpose, one-step mutagenic PCR with Accusure™ DNA Polymerase (Bioline) was used following the manufacturer's instructions. The new plasmid, containing the mutated sequence, is designated as pBTR1-E104C. Expression and purification of both $Cc$ species was carried out as previously described[68]. Briefly, transformed bacteria were incubated

for 20 h at 150 rpm and 30 °C in 2.5 L standard lysogeny broth (LB) rich medium (10 g·L$^{-1}$ tryptone, 5 g·L$^{-1}$ yeast extract and 10 g·L$^{-1}$ NaCl), supplemented with 100 µg·mL$^{-1}$ ampicillin. Cells were harvested via centrifugation at 5000 x g for 10 min at 4 °C, resuspended in 10 mM tricine buffer pH 8.5 supplemented with cOmplete protease inhibitor cocktail (#11836145001, Roche), ruptured by sonication for 5 min, and then centrifuged at 14,000 x g for 60 min. The supernatant was loaded onto a Nuvia S column (#7324720, Biorad) and eluted with a 25 to 500 mM NaCl gradient in 10 mM tricine buffer pH 8.5. Phosphomimetic Y48$p$CMF $Cc$ was expressed using the evolved tRNA synthetase technique, as previously described[41]. Briefly, *E. coli* BL21 (DE3) cells were co-transformed with either pBRTR1-Y48AMBER or pBTR1-Y48AMBER/E104C, and pEVOL/$p$CMF/tRNA plasmids—which contains the unnatural tRNA/aminoacyl-tRNA synthetase pair that recognizes the $p$CMF—and were cultured for 20 h at 150 rpm and 30 °C in M9 minimal medium supplemented with 100 µg·mL$^{-1}$ ampicillin, 20 µg·mL$^{-1}$ chloramphenicol, 16.7 µg·mL$^{-1}$ δ-aminolevulinic acid hydrochloride and 263 µg·mL$^{-1}$ non-canonical amino acid $p$CMF. Protein expression was induced with 1 mM isopropyl-β-D-thiogalactoside and 0.02% arabinose. Cells were harvested by centrifugation and purified similarly as previously described for $Cc$[41]. The soluble domain of *Arabidopsis thaliana* $Cc_1$ was expressed and purified as previously described[12]. Briefly, *E. coli* BL21 (DE3) cells were co-transformed with pET-$Cc_1$ and pEC86 plasmids (which constitutively express the eight cytochrome $c$ maturation genes *ccmABCDEFGH*) to produce the soluble domain of $Cc_1$ (residues 64–265, Gene ID 834081). The cells were grown for 24 h at 150 rpm and 30 °C in 2 L of standard LB rich medium, supplemented with 50 µg·mL$^{-1}$ kanamycin, 12 µg·mL$^{-1}$ chloramphenicol and 16.7 µg·mL$^{-1}$ δ-aminolevulinic acid hydrochloride. Harvested cells were resuspended in 20 mM Tris-HCl buffer pH 8.0 supplemented with cOmplete protease inhibitor cocktail (#11836145001, Roche) and physically ruptured by sonication for 5 min. The supernatant was loaded onto a Nuvia Q column (#7324721, Biorad) and eluted with a 25 to 500 mM NaCl gradient in 20 mM Tris-HCl buffer pH 8.0. Fractions containing $Cc_1$ were further purified by size exclusion with a ENrich SEC 70 column (#7801070, Biorad). All protein samples were dialyzed against 10 mM sodium phosphate buffer at pH 6.5.

### Electrochemical scanning tunneling microscopy-spectroscopy
**Sample preparation.** Atomically flat Au(111) single-crystal disks (10 mm diameter and 1 mm thickness, MaTecK) were flame annealed and electrochemically polished[45]. The Au(111) electrodes were incubated with $Cc_1$ in sodium phosphate buffer 10 mM, pH = 6.5 at a 56 µM concentration. The EC-STM probes (Au wire 0.25 mm in diameter; GoodFellow) were insulated with Apiezon™[45], and were incubated with a 12.5 µM solution of either $Cc$ (E104C) or Y48$p$CMF $Cc$ (E104C), in sodium phosphate buffer 10 mM, pH = 6.5. All incubations were conducted overnight and at 4 °C.

**EC-STM measurements.** All experiments were performed with a PicoSPM microscope head and a PicoStat bipotentiostat (Agilent, USA) controlled by Dulcinea electronics (Nanotec Electronica, Spain) using the WSxM 4.13 software. A custom-made electrochemical liquid cell with a standard sample plate was used in four-electrode configuration with the STM Au(111) sample and probe as working electrodes, a 0.25 mm diameter Pt80/Ir20 wire as counter electrode and a miniaturized ultralow leakage membrane Ag/AgCl (SSC) reference electrode filled with 3 M KCl. The potentials of the Au(111) electrode sample ($U_S$) and EC-STM probe ($U_P$) were expressed against this reference. The electrochemical cell was cleaned with piranha solution (3:1 v/v solution of $H_2SO_4$ and $H_2O_2$) before each measurement. Caution: piranha solution is a strong oxidizer and a strong acid. It should be handled with extreme care, as it reacts violently with most organic materials. The experiments

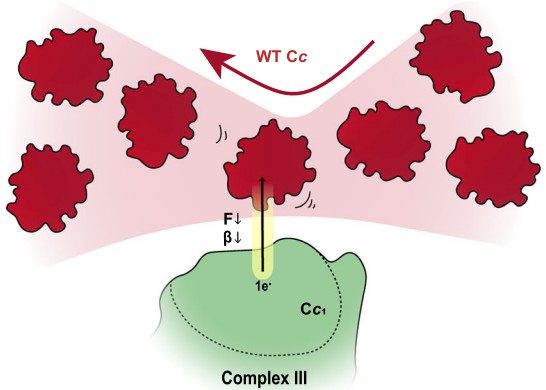
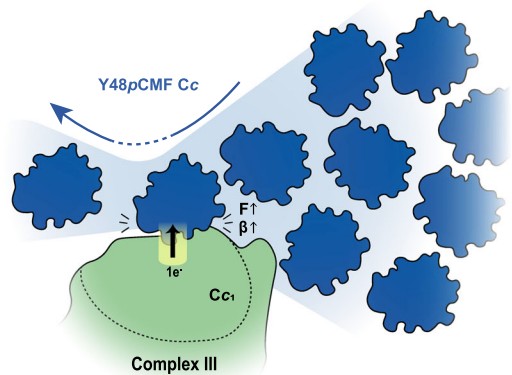

**Fig. 5 | Proposed model showing how phosphorylation of cytochrome $c$ promotes a closer contact with cytochrome $c_1$ and diminishes the turnover of molecules at complex III.** (**Left**) WT cytochrome $c$ (red) can receive electrons from cytochrome $c_1$ of complex III at long distances (low β). Overall turnover of cytochrome $c$ molecules at complex III is high due to reversible binding in equilibrium conditions (low F). (**Right**) Compared to WT C$c$ (red), the dissociation constant ($K_D$) of the Y48$p$CMF variant (blue) is lower and it must get closer to complex III for electron transfer (high β). In this situation, the interaction forces are higher (high F) and thereby the overall turnover of electron carriers is reduced. Conductance of the Y48$p$CMF variant is higher (*right*, thick black arrow), compared to that of the WT species (*left*, thin black arrow). Thus, phosphorylation of cytochrome $c$ disrupts the long-distance electron transfer conduit (marked in yellow) previously observed in the wild-type protein[4].

were performed in sodium phosphate buffer 50 mM (pH = 6.5) electrolyte solution, previously filtered through 0.02 μm diameter sterile membrane (Anotop) and degassed with nitrogen.

The Current versus distance (*I-z*) curves were acquired by setting an initial current set point of 0.4 nA, turning the feedback loop off, and recording the probe current at 12 nm·s⁻¹ during probe retraction along 15 nm (20 samples per point, 1024 points) at a constant bias ($U_{bias} = U_P - U_S$). At least 2 independent experiments were conducted in each case. For each independent experiment, a total of 150 curves (20 samples per point, 1024 points) were recorded. The data were treated with the custom-generated MATLAB code (The MATH-WORKS, Inc.) previously reported[4].

For static blinking experiments, a NI-DAQmx and BNC-2110 Lab-VIEW setup was used for data acquisition, LabVIEW software was used for data treatment. The STM probe is set at a fixed current set point from the sample, and at a constant bias, and the feedback is turned off after a period of mechanical stabilization. Current versus time traces are then recorded in captures of few seconds during several hours. Current blinks are observed in the current transient in the form of telegraphic noise (Fig. 2a). At least 2 independent experiments were conducted in each case. The current is transformed to conductance values using $G = I_{blink}/U_{bias}$. Up to a hundred blinks are used to build 2D-blinking maps.

The data were analyzed with OriginPro 8.5.0 SR1 (OriginLab Corp.). n indicates the sample size in all cases. Errors are indicated as standard deviation of the mean.

**Atomic force microscopy-force spectroscopy**
**Sample preparation.** C$c_1$ was immobilized on atomically flat Au(111) single-crystal disks (10 mm diameter, 1 mm thickness, MaTecK). The Au(111) disks were electrochemically polished[45] and flame annealed, cooled under a nitrogen flow and then incubated with a 56 μM solution of C$c_1$ in sodium phosphate buffer 10 mM, pH 6.5 within a humid environment, overnight and at 4 °C.

**AFM probes preparation.** We used V-shaped Si₃N₄ cantilevers (Bruker AFM Probes, Camarillo, CA) having a nominal spring constant of 0.06 N·m⁻¹ with either silicon (SNL probes) or Au-coated (NPG probes) tips.

Au-coated probes were placed on a PDMS surface and incubated with a 25 μL drop of a 26 μM solution of C$c$ (E104C) or Y48$p$CMF C$c$ (E104C) in sodium phosphate buffer 10 mM, pH 6.5. Incubations were performed at 4 °C overnight in a humid environment.

Silicon probes were first treated for 10 minutes in a UV/Ozone ProCleaner™ (BioForce Nanosciences). Then, the probes were aminofunctionalized in atmosphere of 3-aminopropyl)triethoxysilane (APTES) and triethylamine (TEA). For this, the probes were placed inside the desiccator previously flooded with N₂ gas to remove air and moisture. Two small plastic containers with 45 μL of APTES and 15 μL of TEA, separately, were placed close to the probes (on a clean inert surface) and vacuum was on for 5 min. After 1.5 h, the APTES and TEA container were removed, and vacuum was restored. The probes were left in the desiccator overnight at room temperature. The probes were then rinsed with chloroform (2x) and ethanol (2x) and dried under a gentle N₂ flow. When the tips were not used immediately, they were stored in a desiccator under Ar atmosphere. The AFM probes were then placed in glass wells filled with 0.5 ml solution of 1 mg·mL⁻¹ heterobifunctional PEG ($n = 27$) (Mal-PEG$_{27}$-NHS, $O$-[$N$-(3-maleimidopropionyl)aminoethyl]-$O'$-[3-($N$-succinimidyloxy)−3-oxopropyl]heptacosaethylene glycol, Sigma-Aldrich) and 5 μL TEA in chloroform and left for 2 h. The probes were then rinsed with chloroform (2x), ethanol (2x) and MilliQ water and dried under a gentle N₂ flow. The probes were placed onto a PDMS surface and incubated with a 25 μL drop of a 26 μM solution of C$c$ (E104C) or Y48$p$CMF C$c$ (E104C) in sodium phosphate buffer 10 mM, pH 6.5. Incubations were performed at 4 °C overnight in a humid environment.

**AFM-FS measurements.** Force-separation curves were measured with either an MFP3D AFM (Asylum Research, Oxford Instruments) or a NanoWizard 3 BioScience AFM (JPK Instruments, Bruker Nano GmbH) at room temperature and under liquid environment (sodium phosphate buffer 50 mM, pH 6.5). After having measured the sensitivity (V·m⁻¹), the cantilever spring constants were individually calibrated by using the equipartition theorem (thermal noise routine)[69]. Force-separation curves were recorded by approaching and retracting the AFM tip at constant velocity (0.5 to 4 μm·s⁻¹) and in the force map mode over an area of 3 × 3 μm². The maximum applied (contact) force in each cycle was set to 300 pN. At least two independent experiments were conducted in each case. AFM data were acquired and treated using the AFM software and analyzed with OriginPro 8.5.0 SR1 (OriginLab Corp.). Errors are indicated as the standard deviation of the mean.

## Surface plasmon resonance SPRi

C$c$ interaction to C$c_1$ was assayed by SPR using a CS SPRi-Biochip™ (HORIBA Scientific) and a SPRi-Plex II™ system (HORIBA Scientific), as previously described with minor modifications[70]. Briefly, C$c_1$ direct amine immobilization onto the biochip was achieved by deposition of 10 and 15 μM solution drops on the surface and rabbit serum albumin (Sigma-Aldrich) was used as control at reference spots. The binding assays were performed at 298 K in 10 mM sodium phosphate buffer, pH 6.5. WT or Y48$p$CMF C$c$ were flowed at various concentrations (from 2.5 to 10 μM) over the modified surface with C$c_1$ at 50 μL·min⁻¹ rate. The signals from the reference spot surface were subtracted in every sensogram. The data were analyzed and fitted to a one-to-one binding model adapted from Bowles $et$ $al$[71]. with Origin 2018 (OriginLab Corporation).

## Cytochrome $bc_1$ complex activity

The ability of WT and Y48$p$CMF C$c$ variants to act as electron acceptor of cytochrome $bc_1$ complex (complex III) was tested with mammalian mitochondria extracted from HeLa cells. HeLa cells were cultured in Dulbecco's modified Eagle's medium (DMEM; Sigma-Aldrich) supplemented with 10% heat-inactivated fetal bovine serum (FBS; Sigma-Aldrich), 2 mM L-glutamine (Sigma-Aldrich), 100 U·mL⁻¹ streptomycin (Sigma-Aldrich) and 100 μg·mL⁻¹ penicillin (Sigma-Aldrich). Mitochondria were isolated from HeLa cells as previously described and stored at −80 °C in 10 mM Tris-MOPS, 1 mM EGTA/Tris, 200 mM sucrose pH 7.4[72]. Complex III activity was measured spectrophotometrically (Varioskan™ LUX microplate reader, Thermo Scientific™) using the commercial mitochondrial complex III activity kit from Abcam (ab287844), according to the manufacturer's instructions. Net complex III activity was calculated by comparing the C$c$ reduction in the presence and absence of Antimycin A, a complex III specific inhibitor.

## Molecular dynamics simulations

The initial model of the C$c$ and C$c_1$ proteins was done essentially as described by Lagunas et al.[4], built from the PDB structure 3CX5[73], using the Multiseq plugin[74] of the VMD program[75]. The two proteins were separated 3.2 nm along one axis, taking the most external hydrogen atom from the -CH$_3$ distal pairs from each heme group, that is 4.2 nm taking the Fe-to-Fe distance. The molecular dynamics (MD) simulation was set up with the program LEaP included in the Amber suite of programs[76] and the ff99SB protein force field. The titratable residues were modeled in the corresponding protonation state at pH 6.5, which was further checked by analysis of their intermolecular interactions. The AMBER parameters database (http://research.bmh.manchester.ac.uk/bryce/amber) was used for the heme parameters, considering RESP charges for the reduced and oxidized cases from Autenrieth et al[77]. Values from the same reference were used to parametrize the charges of the interacting residues with the heme groups, His18/44 and Met80/166 from C$c$/C$c_1$ respectively. Explicit TIP3P water molecules were used for solvation[78], and the system was neutralized with 33/35 sodium/chlorine atoms, in addition to the 33 NaCl molecules to reproduce a concentration of 50 mM. The final system contained 105541 atoms. Amber14[79] was used for the MD simulations. A restraint on the residues 1–15 for C$c_1$ and 97–104 for C$c$ was applied to model the experimental situation of Cys10 and Cys104, respectively, immobilized on the gold surfaces. A thermal equilibration to 300 K was first done, followed by the equilibration dynamics in the NPT ensemble for 36 ns. We used the SHAKE algorithm, with an integration time step of 2 fs. To obtain the equipotential surfaces, APBS electrostatics calculations[80] were done in the postprocessing step. Average ions and water concentration maps and number density prism volumes were obtained with the Volmap Plugin 1.1. Matplotlib[81] and VMD[75] were used to draw the figures.

## Brownian dynamics simulations

Brownian dynamics (BD) computations were carried out and analyzed using the SDA-flex 7.1 software package[82]. The force-field grids used in BD included all electrostatics and desolvation[65,83] grids. Charges were obtained from PQR files extracted from each MD trajectory. Diffusion constants were computed using the script ARO[84], in the VMD tcl-tk console. Electrostatic grids were generated for every conformer with APBS 3.0[85]. All simulations were carried out at 100 mM ionic strength. In total, 20 structures of C$c_1$ and other 20 of C$c$ were used as input. For $k_{on}$ computations, each of the 20 C$c_1$ molecules were treated as targets in separate computations and set in the coordinate origin, whereas the 20 conformations of C$c$ were used as input for Monte-Carlo conformation exchange during the simulation of their diffusion. Conformational exchange was allowed every 2.5 ns. A total of 200,000 diffusion trajectories (20 × 10,000) were then computed. Origin 2019b (Originlab) was used for statistical analysis and data representation.

## Reporting summary

Further information on research design is available in the Nature Portfolio Reporting Summary linked to this article.

## Data availability

All the data generated in this study are available within the main text and the Supplementary Information file; source data are provided in the Source Data file. Data is also available from the corresponding author upon request. Source data are provided with this paper.

## Code availability

The source code used to compute the presented results can be obtained from the electronic supplementary material available at: https://www.nature.com/articles/s41467-018-07499-x (reference 4).

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

## Acknowledgements

This research received funding from the European Union Research and Innovation Programme Horizon 2020 – HBP SG3 (945539) to P.G., DEEPER (ICT-36-2020-101016787) to P.G., Ministry of Science and Innovation (Grants PID2021-126663NB-I00 to I.D.-M., PID2019-111493RB-I00 to P.G., PGC2018-096049-B-I00 to I.D.-M., CTQ2015-66194-R to A.L. and M.I.G., and PID2020-118893GB-I00 to C.R.), the Spanish Structures of Excellence María de Maeztu (MDM-2017-0767 to C.R.), Fonds Européen de Développement Économique et Régional (FEDER) funds, Agency for Management of University and Research Grants/Generalitat de Catalunya (CERCA Programme, (2017-SGR-1079 to J.S., 2017-SGR-1442 to P.G. and 2017-SGR-1189 to C.R.), Andalusian Government (BIO-198 to I.D.-M., US-1254317 to I.D.-M., US-1257019 to M.A.R., P18-FR-3487 to I.D.-M., and P18-HO-4091 to I.D.-M., US/JUNTA/FEDER, UE), University of Seville (VI PPIT to I.D.-M.) and the Ramón Areces Foundation (2021-2024 to I.D.-M.).

The project CECH, 001-P-001682 to P.G. is co-financed by the European Union Regional Development Fund within the framework of the ERDF Operational Program of Catalonia 2014-2020 with a grant of 50% of total eligible cost. G.P.-M. was awarded a PhD fellowship from the Spanish Ministry of Education, Culture and Sport (FPU17/04604). A.L., M.I.G., J.S., and P.G. were supported by the Biomedical Research Networking Center (CIBER), Spain (CB06/01/0055 and CB06/01/0081). CIBER is an initiative funded by the VI National R&D&i Plan 2008-2011, Iniciativa Ingenio 2010, Consolider Program, CIBER Actions, and the Instituto de Salud Carlos III, with the support of the European Regional Development Fund (ERDF). This work was supported by grants. The authors also acknowledge Dr. Albert Cortijos Aragonès and Jessica Sierra Agudelo for help with the EC-STM blinking measurements and data analysis, and Manuel López Ortíz for technical assistance. We also thankfully acknowledge the computer resources at CTE-Power and the technical support provided by the Red Española de Supercomputación-Barcelona Supercomputing Center.

## Author contributions

P.G. and A.L. designed the ECSTM experiments and A.L. performed the ECSTM experiments and data analyses; M.I.G. designed the AFM-FS experiments and M.I.G., L.C.-F. and S.O.-T. performed the AFM-FS experiments and data analyses; I.D.-M. and M.A. de la R. discussed the experiments and designed protein production, SPRi and Complex III activity experiments and A.G.-C. and G.P.-M. produced and purified the proteins, G.P.-M. performed the SPRi and Complex III activity experiments; P.G. and C.R. designed the MD experiments, and A.M.J.G. and A.N.-H. performed the simulations. A.D.-Q. performed the Brownian dynamics simulations. A.L, M.I.G., I.D.-M. and P.G. wrote the paper with contributions from A.M.J.G., G.P.-M, A.N.-H., A.G.-C., M.A. de la R., C.R., and J.S.; M.I.G. and A.L. contributed equally.

## Competing interests

The authors declare no competing interests.
