## [Peer Review File · Nature Communications]

Phosphorylation disrupts long-distance electron transport in cytochrome cEditorial Note: This manuscript has been previously reviewed at another journal that is not operating a transparent peer review scheme. This document only contains reviewer comments and rebuttal letters for versions considered at Nature Communications.

REVIEWER COMMENTS

Reviewer #2 (Remarks to the Author):

My previous concerns have been satisfactorily addressed by the authors. I am glad to recommend publication of the revised manuscript in Nature Communications.

Reviewer #4 (Remarks to the Author):

The manuscript describes the interaction between complex III and its physiological partner-the intermembrane mitochondrial cytochrome c (Cc), either in its phosphorylated (Tyr 48) or non-phosphorylated state. Some of the authors previously describe long distance electron transfer (Lagunas et al., 2018, Nat. Comm 5157) between Cc and complex III with the same system. Now, they combine EC-STM, AFM, protein-protein interaction, enzymatic assays and MD simulations to characterize electron transfer with the phosphorylated Cc.

The results show lost on long distance electron transfer, and attribute this loss to a strong affinity between the two proteins when Cc is phosphorylated. These data are relevant for the research field considering Cc role inside the cell, as well as the different pathways in which it intervenes, taking place inside or outside the mitochondria.

The system uses the terminal redox center from Arabidopsis thaliana complex III- the cytochrome c1 (Cc1) soluble domain and a p-carboxy-methyl-phenylalanine (pCMF) modification to mimic Tyr phosphorylation rather than replacement of the Tyr by a glutamate residue. The rationale for the choice of this mimicking strategy is clearly stated in manuscript introduction. References were provided for Cc1 production, but a quick search indicated another reference instead of the methods or the specific domain produced. To avoid endless search for the published method within the references, a brief description on Cc1 production can be included in the methods, mentioning the specific length of the truncated domain produced.

Concerning Cc and pCMF-Cc, the authors present electrochemical data (supplementary figure S2) that show no significant shift in the midpoint redox potential from these two proteins (approximately 10 mV). However, physicochemical characterization of the pCMF-Cc performed by some of the authors and referenced in the manuscript (Guerra-Castellano, A., et. al. 2015, Ref 43) give a lower midpoint redox potential on the phosphorylated Cc (50 mV lower) in the physiologic pH range. Can the authors clarify this and suggest the lack on shifts in the pCMF-Cc redox potential. Additionally, supplementary figure S1 show pCMF-Cc with a higher heme solvent accessibility, without changes in the axial coordination (as expected), this solvent exposure would be expected to promote shifts in the heme-Fe environment, producing shifts in the midpoint redox potential.

Enzymatic assays were performed to quantify complex III activity with Cc and pCMF-Cc. Do the authors consider experiments to measure the production of ROS by Complex III with Cc and pCMF-Cc ? This information would bring relevant information to the presented work.

During the manuscript, authors use the termination active site to describe either the Cc1 or the Cc heme. Since both proteins are electron transfer proteins/domains, I suggest as an alternative redox centers or redox active centers/sites like used by the authors in some of the manuscript sentences

minor suggestions:

Figures present low quality, especially figures 2a, 2b, 2d, 3a, 3b, 4i;

Figure 3B: x-axis label overlaps graphic scale;

Line 264: figure 2 caption , clarify n values;

Line 319: contour maps Fr vs. Ir, Fig. 3b). Should it be figure 3d

Line 490: Mutated. The gene sequence was mutated, but in the produced protein the same aminoacid

is there with a modified pCMF, therefore I suggest modified;

Table 1: the kon constants values should present error value with the same notation as the value, similar to koff and the constants presented in table 2

Line 436: Figure 4 caption (M, Mol-1L-1) both mean the same thing, is it necessary ?

Line 518: Cc Y48pCMF mutant. Same as above, no mutation was introduced rather a modification that mimics a post translational modification

Reviewer #5 (Remarks to the Author):

This manuscript is well planned, established and written. Additionally, this is significantly revised with responding to the other reviewers' comments. I believe this should be accepted.

I should note one thing. In the manuscript, the authors uses the word of "we" as a subject, for example L305 in P8. In these scientific manuscripts, we never use "we" as a subject. Instead of personal words, the inanimate subject including "it" or "that" should be used to keep objectivity. The authors should revise these points in the manuscript before acceptance.

Reply to the reviewer's comments

REVIEWER COMMENTS

Reviewer #2 (Remarks to the Author):

My previous concerns have been satisfactorily addressed by the authors. I am glad to recommend publication of the revised manuscript in Nature Communications.

The authors thank the reviewer for his/her approval for publication.

Reviewer #4 (Remarks to the Author):

The manuscript describes the interaction between complex III and its physiological partner-the intermembrane mitochondrial cytochrome *c* (*Cc*), either in its phosphorylated (Tyr 48) or non-phosphorylated state. Some of the authors previously describe long distance electron transfer (Lagunas et al., 2018, Nat. Comm 5157) between *Cc* and complex III with the same system. Now, they combine EC-STM, AFM, protein-protein interaction, enzymatic assays and MD simulations to characterize electron transfer with the phosphorylated *Cc*. The results show loss on long distance electron transfer and attribute this loss to a strong affinity between the two proteins when *Cc* is phosphorylated. These data are relevant for the research field considering *Cc* role inside the cell, as well as the different pathways in which it intervenes, taking place inside or outside the mitochondria.

We highly appreciate the reviewer's opinion about the manuscript.

The system uses the terminal redox center from *Arabidopsis thaliana* complex III- the cytochrome *c*₁ (*Cc*₁) soluble domain and a *p*-carboxy-methyl-phenylalanine (*p*CMF) modification to mimic Tyr phosphorylation rather than replacement of the Tyr by a glutamate residue. The rationale for the choice of this mimicking strategy is clearly stated in manuscript introduction. References were provided for *Cc*₁ production, but a quick search indicated another reference instead of the methods or the specific domain produced. To avoid endless search for the published method within the references, a brief description on *Cc*₁ production can be included in the methods, mentioning the specific length of the truncated domain produced.

Following reviewer's suggestion, we have included a brief description of *Cc*₁ production in the main text and the Methods section (page 4, lines 154-156 and pages 15-16, lines 568-574).

Concerning *Cc* and *p*CMF-*Cc*, the authors present electrochemical data (supplementary figure S2) that show no significant shift in the midpoint redox potential from these two proteins (approximately 10 mV). However, physicochemical characterization of the *p*CMF-*Cc* performed by some of the authors and referenced in the manuscript (Guerra-Castellano, A., et. al. 2015, Ref 43) give a lower midpoint redox potential on the phosphorylated *Cc* (50 mV lower) in the physiologic pH range. Can the authors clarify this and suggest the lack on shifts in the *p*CMF-*Cc* redox potential. Additionally, supplementary figure S1 show *p*CMF-*Cc* with a higher heme solvent accessibility, without changes in the axial coordination (as expected), this solvent exposure would be expected to promote shifts in the heme-Fe environment, producing shifts in the midpoint redox potential.

As the reviewer pointed out, the Y48*p*CMF *Cc* variant displays a ~50 mV lower midpoint redox potential value compared to the WT protein *in solution* (Guerra-Castellano, et al. 2015) and immobilized electrostatically onto gold electrodes modified with negatively-charged self-assembled monolayers (SAM) of different nature (Olloqui-Sariego, et al. 2022). In the last scenario, the heme group of *Cc* would be close to the electrode due to the positively charged patch of lysine residues that surrounds the heme cleft. The higher heme solvent accessibility and protein dynamics of the Y48*p*CMF variant further modulates *Cc* orientation and binding to the electrode, as well as its midpoint potential (Olloqui-Sariego 2022). The differences in midpoint redox

potential between WT and Y48 p CMF species are increased upon immobilization onto SAMs with stronger local electric field, suggesting that the binding-triggered changeover on the redox properties of the Y48 p CMF variant arises from a subtle distortion of the active site, likely induced by alteration of the hydrogen-bonding network and electrostatics of the heme crevice rather than by coarse structural changes. Therefore, in the case of SAM-mediated and unspecific attachment via interactions with the protein hydrophobic patch, changes on the hydrophobicity of the patch (for example the introduction of charges due to phosphorylation) are likely to produce important changes in binding and orientation and, therefore, redox potential.

However, the difference in midpoint redox potential of our covalently bound *Cc* mutants (E104C and Y48 p CMF/E104C) is smaller (~10 mV). Since the orientation of both proteins with respect to the electrode is identical as imposed by the immobilization through the C-terminal cysteine residue, this reduced the effect of charges on the midpoint potential. In this situation, the minor change observed in the midpoint redox potential value could be ascribed to the change in the net charge of the protein as described by Petrović et al. (Petrović, et al. 2005).

Enzymatic assays were performed to quantify complex III activity with *Cc* and p CMF-*Cc*. Do the authors consider experiments to measure the production of ROS by Complex III with *Cc* and p CMF-*Cc* ? This information would bring relevant information to the presented work.

We thank the reviewer for the suggestion of this exciting experiment.

Complex I and complex III are the major sites for ROS production in the electron transport chain. Extensive work has been done over the last years to understand and localize the site of production of ROS within the mitochondrial respiratory complexes. Measurements of mitochondrial ROS production in intact living cells are usually performed using fluorescent probes such as MitoSOX. Indeed, the effect of Thr/Ser phosphomimetic variants of *Cc* has been analyzed *in cell* using this approach (Wan, et al. 2019; Kalpage, et al. 2020). In these articles, the authors transfected *Cc* double-knockout mouse lung fibroblast cell line with plasmids encoding for phosphomimetic variants of *Cc* (Ser/Thr-to-Glu mutants) and they found out that cells expressing phosphomimetic variants have a lower mitochondrial potential that corresponds with lower mitochondrial ROS production. Even though such an approach is elegant and useful to predict the effect of the *in vivo* phosphorylation, it is not applicable to study the effect of *Cc* tyrosine phosphorylation. The non-canonical Tyr-to- p CMF phosphomimetic for *Cc* Tyr48 phosphorylation has been proven to be a better alternative to the traditional Tyr-to-Glu phosphomimetic variant. The lack of a method to introduce such non-canonical amino acid (p CMF) in mitochondrial proteins in mammalian cell cultures hampers analyzing the effect of Tyr48 phosphorylation in mitochondrial ROS production in the cell.

Another approach commonly used to evaluate the ROS production by mitochondria and submitochondrial particles is based on enzyme-linked fluorescent techniques where Amplex Red reagent in the presence of horseradish peroxidase reacts with H₂O₂ in a 1:1 stoichiometry to produce highly fluorescent resorufin (Chen, et al. 2003). In order to test the effect of p CMF variant of *Cc*, we would need to add exogenous *Cc* to the mitochondrial extract. *Cc* is able to scavenge superoxide and hydrogen peroxide due to its superoxide-oxidizing and pseudo-peroxidase activity (Butler, et al. 1975; Korshunov, et al. 1999; Pasdois, et al. 2011). In this context, Y48 p CMF *Cc* variant displays an increased peroxidase and ROS scavenger activity compared to WT (Moreno-Beltrán, et al. 2017). In the present work, we demonstrated that *Cc* phosphorylation slows down complex III activity. However, the effect on ROS production by complex III deceleration is difficult to predict and measure. *In vitro* measurements with isolated complex III showed that superoxide is only produce when the electron flow is severely hampered, e.g., by blocking Qi center with antimycin A (Borek, et al. 2008). Some authors have proposed that superoxide generation associated with blockage of Qi site of complex III can be enhanced by blocking the electron outflow through *Cc*₁, either by mutation (Sarewicz, et al. 2010) or by increasing the ionic strength enough (400 mM NaCl) to inhibit the interaction between *Cc*₁ and *Cc* (Sarewicz, et al. 2008). In our case, complex III activity is not completely blocked; therefore, one can expect that the increase in the amount of ROS generated will be low. Slowdown of CIII activity will favor

the oxidized pool of Cc molecules in the intermembrane space. This would increase the ability of the Cc pool to oxidize superoxide, either reducing the total ROS production or buffering the increase mediated by the reduced complex III activity (Pasdois 2011). Moreover, the exogenous Y48pCMF would be able to scavenge more efficiently the hydrogen peroxide generated during the Amplex Red assay, making the interpretation of the results nontrivial.

For all these reasons, the investigation of the effect of Tyr48 phosphorylation on the ROS production by complex III is very relevant but would require an extensive analysis to identify each of the individual variables controlling it. We are considering these directions in a follow-up project. Nevertheless, we thank the reviewer for bringing up such relevant suggestion.

During the manuscript, authors use the termination active site to describe either the Cc₁ or the Cc heme. Since both proteins are electron transfer proteins/domains, I suggest as an alternative redox centers or redox active centers/sites like used by the authors in some of the manuscript sentences

Following the reviewer's suggestion, "active site" has been changed to "redox active site" throughout the manuscript.

minor suggestions:

Figures present low quality, especially figures 2a, 2b, 2d, 3a, 3b, 4i;

The quality of the figures has been improved as requested, and high-resolution files for all figures will be uploaded during resubmission.

Figure 3B: x-axis label overlaps graphic scale;

The x-axis of Fig. 3b has been corrected.

Line 264: figure 2 caption, clarify n values;

"n" corresponds to the sample size in all cases. For the two-dimensional histograms of blink current (I_{blink}) and blink lifetime In Figure 2b, "n" corresponds to the total conductance values experimentally determined and used to construct the histograms. It has been indicated in the corresponding Figure caption (Lines 258-260, page 7) and in the Methods section (Lines 610-611, page 16), respectively, and highlighted in the revised version of the manuscript.

Line 319: contour maps Fr vs. lr, Fig. 3b). Should it be figure 3d

We have corrected this mistake.

Line 490: Mutated. The gene sequence was mutated, but in the produced protein the same aminoacid is there with a modified pCMF, therefore I suggest modified;

"mutated with pCMF" has been changed to "herein mimicked by replacement with pCMF"

Table 1: the kon constants values should present error value with the same notation as the value, similar to koff and the constants presented in table 2

We have revised the error value notation in Table 2.

Line 436: Figure 4 caption (M, Mol-1L-1) both mean the same thing, is it necessary ?

We have removed Mol⁻¹ L⁻¹.

Line 518: Cc Y48pCMF mutant. Same as above, no mutation was introduced rather a modification that mimics a post translational modification

"Y48pCMF mutant" has been changed to "Y48pCMF variant"

Reviewer #5 (Remarks to the Author):

This manuscript is well planned, established and written. Additionally, this is significantly revised with responding to the other reviewers' comments. I believe this should be accepted.

We thank the reviewer for his/her positive comments and approval for publication.

I should note one thing. In the manuscript, the authors uses the word of “we” as a subject, for example L305 in P8. In these scientific manuscripts, we never use “we” as a subject. Instead of personal words, the inanimate subject including “it” or “that” should be used to keep objectivity. The authors should revise these points in the manuscript before acceptance.

Following the recommendations from Nature on ‘How to construct a *Nature* summary paragraph’, we choose to use the personal object “we” instead of the impersonal object “it”.

References

- Borek, Sarewicz, and Osyczka. 2008. “Movement of the iron-sulfur head domain of cytochrome bc1 transiently opens the catalytic qo site for reaction with oxygen.” *Biochemistry* 47 (47): 12365–70. <https://doi.org/10.1021/bi801207f>.
- Butler, Jayson, and Swallow. 1975. “The reaction between the superoxide anion radical and cytochrome c.” *BBA - Bioenerg.* 408 (3): 215–22. [https://doi.org/10.1016/0005-2728\(75\)90124-3](https://doi.org/10.1016/0005-2728(75)90124-3).
- Chen, Vazquez, Moghaddas, Hoppel, and Lesnefsky. 2003. “Production of reactive oxygen species by mitochondria: central role of complex iii.” *J. Biol. Chem.* 278 (38): 36027–31. <https://doi.org/https://doi.org/10.1074/jbc.M304854200>.
- Guerra-Castellano, Díaz-Quintana, Moreno-Beltrán, López-Prados, Nieto, Meister, Staffa, et al. 2015. “Mimicking tyrosine phosphorylation in human cytochrome c by the evolved trna synthetase technique.” *Chem. – A Eur. J.* 21 (42): 15004–12. <https://doi.org/https://doi.org/10.1002/chem.201502019>.
- Kalpage, Wan, Morse, Lee, and Hüttemann. 2020. “Brain-specific serine-47 modification of cytochrome c regulates cytochrome c oxidase activity attenuating ros production and cell death: implications for ischemia/reperfusion injury and akt signaling.” *Cells* 9 (8). <https://doi.org/10.3390/cells9081843>.
- Korshunov, Krasnikov, Pereverzev, and Skulachev. 1999. “The antioxidant functions of cytochrome c.” *FEBS Lett.* 462 (1–2): 192–98. [https://doi.org/10.1016/S0014-5793\(99\)01525-2](https://doi.org/10.1016/S0014-5793(99)01525-2).
- Moreno-Beltrán, Guerra-Castellano, Díaz-Quintana, Del Conte, García-Mauriño, Díaz-Moreno, González-Arzola, et al. 2017. “Structural basis of mitochondrial dysfunction in response to cytochrome c phosphorylation at tyrosine 48.” *Proc. Natl. Acad. Sci.* 114 (15): E3041–50. <https://doi.org/10.1073/pnas.1618008114>.
- Ollolqui-Sariego, Pérez-Mejías, Márquez, Guerra-Castellano, Calvente, De la Rosa, Andreu, and Díaz-Moreno. 2022. “Electric field-induced functional changes in electrode-immobilized mutant species of human cytochrome c.” *Biochim. Biophys. Acta - Bioenerg.* 1863 (7): 148570. <https://doi.org/10.1016/j.bbabi.2022.148570>.
- Pasdois, Parker, Griffiths, and Halestrap. 2011. “The role of oxidized cytochrome c in regulating mitochondrial reactive oxygen species production and its perturbation in ischaemia.” *Biochem. J.* 436 (2): 493–505. <https://doi.org/10.1042/BJ20101957>.
- Petrović, Clark, Yue, Waldeck, and Bowden. 2005. “Impact of surface immobilization and solution ionic strength on the formal potential of immobilized cytochrome c.” *Langmuir* 21

(14): 6308–16. <https://doi.org/10.1021/la0500373>.

Sarewicz, Borek, Cieluch, Świerczek, and Osyczka. 2010. “Discrimination between two possible reaction sequences that create potential risk of generation of deleterious radicals by cytochrome bc₁. implications for the mechanism of superoxide production.” *Biochim. Biophys. Acta - Bioenerg.* 1797 (11): 1820–27. <https://doi.org/10.1016/j.bbabi.2010.07.005>.

Sarewicz, Borek, Daldal, Froncisz, and Osyczka. 2008. “Demonstration of short-lived complexes of cytochrome c with cytochrome bc₁ by epr spectroscopy: implications for the mechanism of interprotein electron transfer.” *J. Biol. Chem.* 283 (36): 24826–36. <https://doi.org/10.1074/jbc.M802174200>.

Wan, Kalpage, Vaishnav, Liu, Lee, Mahapatra, Turner, et al. 2019. “Regulation of respiration and apoptosis by cytochrome c threonine 58 phosphorylation.” *Sci. Rep.* 9 (1): 15815. <https://doi.org/10.1038/s41598-019-52101-z>.

REVIEWERS' COMMENTS

Reviewer #4 (Remarks to the Author):

The authors in the reply to reviewer's comments satisfactorily addressed all my concerns, and the manuscript was emend accordingly.

I recommend the publication of the revised manuscript.